# Validation of asthma recording in the Clinical Practice Research Datalink (CPRD)

Francis Nissen,[1] Daniel R Morales,[2] Hana Mullerova,[3] Liam Smeeth,[1] Ian J Douglas,[1] Jennifer K Quint[4]

[1]Department of Non-Communicable Disease Epidemiology, London School of Hygiene and Tropical Medicine, London, UK
[2]Division of Population Health Sciences, University of Dundee, Dundee, UK
[3]RWE & Epidemiology, GSK R&D, Uxbridge, UK
[4]National Heart and Lung Institute, Imperial College, London, UK

**Correspondence to**
Dr Francis Nissen;
francis.nissen@lshtm.ac.uk

## ABSTRACT

**Objectives** The optimal method of identifying people with asthma from electronic health records in primary care is not known. The aim of this study is to determine the positive predictive value (PPV) of different algorithms using clinical codes and prescription data to identify people with asthma in the United Kingdom Clinical Practice Research Datalink (CPRD).

**Methods** 684 participants registered with a general practitioner (GP) practice contributing to CPRD between 1 December 2013 and 30 November 2015 were selected according to one of eight predefined potential asthma identification algorithms. A questionnaire was sent to the GPs to confirm asthma status and provide additional information to support an asthma diagnosis. Two study physicians independently reviewed and adjudicated the questionnaires and additional information to form a gold standard for asthma diagnosis. The PPV was calculated for each algorithm.

**Results** 684 questionnaires were sent, of which 494 (72%) were returned and 475 (69%) were complete and analysed. All five algorithms including a specific Read code indicating asthma or non-specific Read code accompanied by additional conditions performed well. The PPV for asthma diagnosis using only a specific asthma code was 86.4% (95% CI 77.4% to 95.4%). Extra information on asthma medication prescription (PPV 83.3%), evidence of reversibility testing (PPV 86.0%) or a combination of all three selection criteria (PPV 86.4%) did not result in a higher PPV. The algorithm using non-specific asthma codes, information on reversibility testing and respiratory medication use scored highest (PPV 90.7%, 95% CI 82.8% to 98.7%), but had a much lower identifiable population. Algorithms based on asthma symptom codes had low PPVs (43.1% to 57.8%)%).

**Conclusions** People with asthma can be accurately identified from UK primary care records using specific Read codes. The inclusion of spirometry or asthma medications in the algorithm did not clearly improve accuracy.

**Ethics and dissemination** The protocol for this research was approved by the Independent Scientific Advisory Committee (ISAC) for MHRA Database Research (protocol number15_257) and the approved protocol was made available to the journal and reviewers during peer review. Generic ethical approval for observational research using the CPRD with approval from ISAC has been granted by a Health Research Authority Research Ethics Committee

## Strengths and limitations of this study

► This study describes algorithms to identify people with asthma from Clinical Practice Research Datalink, a large electronic health records database, and measures the positive predictive value of those algorithms.
► Supporting information, including outpatient referral letters, other emergency department discharge letters, airflow measurements and radiography records were used to identify patients with asthma and calculate the test measures.
► The gold standard to calculate a positive predictive value (general practitioner (GP) questionnaire and review by study physicians) is not absolute, even though information from secondary care was used.
► GPs of patients with complicated medical histories could be less likely to return the questionnaire, but remuneration makes this less likely.

(East Midlands—Derby, REC reference number 05/MRE04/87). The results will be submitted for publication and will be disseminated through research conferences and peer-reviewed journals.

## BACKGROUND

Asthma is one of the most common chronic diseases, with an estimated prevalence of 241 million people worldwide with asthma.[1] The UK has one of the highest asthma prevalence and mortality rates in Europe.[2 3] The disease is a significant burden to the National Health Service, with 5.4 million people receiving treatment and approximately 65 000 hospital admissions yearly.[4] Cough, wheeze, breathlessness and chest tightness are its core symptoms[5] but it has a wide variety of different presentations.[6]

Electronic health records (EHR) have been adopted worldwide, facilitating the construction of large population-based patient databases that have become available over the last decades for epidemiological research.[7] Validation of diagnoses or outcomes based on

codes recorded in EHRs is required because their accuracy is uncertain, and this may affect the reliability and validity of subsequent observational studies. The quality of studies generated from EHRs may be debatable unless their data are validated for specific research purposes.[8–11]

The diagnosis of asthma relies on clinical judgement based on a combination of patient history, physical examination and confirmation of the variability or reversibility of airflow obstruction using airflow measurements. This can make it difficult to assess the accuracy of asthma diagnoses in EHR-based epidemiological studies as some symptoms and airflow measurements may not be recorded. In addition, individuals affected by asthma can vary greatly in their presentation and symptoms are sometimes similar to other respiratory diseases such as chronic obstructive pulmonary disease (COPD).[12 13]

The aim of this study was to test the accuracy of different approaches to identifying asthma in the UK Clinical Practice Research Datalink (CPRD) using the positive predictive value (PPV), by comparing the database records with a gold standard constructed from a review by two study physicians based on information provided by general practitioners (GPs) of patients with asthma.

## METHODS
### Dataset
The CPRD is a large UK primary care database containing anonymised data on the people registered with primary care practices from across the UK. CPRD is representative of the UK population with regard to age and sex.[14 15] Within CPRD, diagnostic accuracy has been demonstrated to be high for many conditions and diseases, including COPD.[16–19] CPRD contains detailed clinical information on diagnoses, prescriptions, laboratory tests, symptoms and hospital referrals, in addition to basic sociodemographic information recorded by the GP. These GPs act as primary care providers and gatekeepers for other National Health Service services, and information from other healthcare providers is also transmitted back to the GP. Clinical events and diagnoses are coded as Read codes, a dictionary of clinical terms widely used in the UK National Health Services by both primary and secondary healthcare providers. Validation studies aid to ensure credibility and quality of epidemiological studies done in CPRD.[10]

### Inclusion criteria
The study population consisted of people who had a record for a Read code indicating possible asthma in the 2 years before the index date (1 December 2015) and who were registered in a GP practice meeting CPRD quality criteria. The Read code list is included in the supplementary appendix 1 and is available online at http://datacompass.lshtm.ac.uk/236/.%20The data collection was planned before the index test and reference standard were performed. This timespan was chosen for several

reasons: to overcome potential changes in quality of asthma diagnosis and recording over time; to reduce the chance that the database records were out of date; and to ensure the medical records were still available to GPs. People were identified at random based on one of eight predefined algorithms exclusively, which means that we populated the algorithm resulting in the smallest population first and subsequently removed these people from the cohort, to prevent them from also being selected for another algorithm. We randomly selected 800 possible asthma cases for validation. Of these, 116 asthma cases were excluded because their GPs no longer participated with CPRD at the time questionnaires were sent to the clinicians for validation, as shown in figure 1. Because of the changes in CPRD data governance after the start of the study it was not possible to select replacement patients.

### GP questionnaire
CPRD mailed a two-page questionnaire to the GPs of the people selected for inclusion as described above, requesting confirmation of current asthma diagnosis and additional information to support this diagnosis. This questionnaire can be found in online supplementary appendix 2. The questionnaire was designed to ascertain the diagnosis of asthma and verify the date of diagnosis. The questions included evidence of reversible airway obstruction, current symptoms, smoking history, respiratory comorbidities and Quality Outcome Framework (QOF) indicators. QOF is a national financial incentive scheme for GPs in the UK encouraging regular disease indicator measurement and recording. Asthma is one of the included diseases, and its indicators including airflow measurements and interference with work and night's rest.[20]

Specific information available from the medical record including spirometry printouts and hospital respiratory outpatient letters were also requested. Data were encrypted twice to ensure anonymity, between practices and CPRD and also from CPRD to researchers. A questionnaire was considered invalid if it was returned blank or every question was answered 'unknown'.

### Code lists and algorithms
Lists of medical codes (Read codes) deemed as specific and non-specific for asthma based on study physicians' opinion were created prior to the start of the study. Read codes are a hierarchical clinical coding system that are used in general practice in the UK and are entered by the GP into a computer program called Vision. Each Read code is linked to a specific string of text, which refers to a single diagnosis or symptom. These data are then uploaded by CPRD after they have been processed and quality checked. The list of codes used for specific or definite asthma codes and non-specific or probable asthma codes can be found in online supplementary appendix 1.

Combinations of Read code lists, evidence of reversibility testing and respiratory medication use were used to

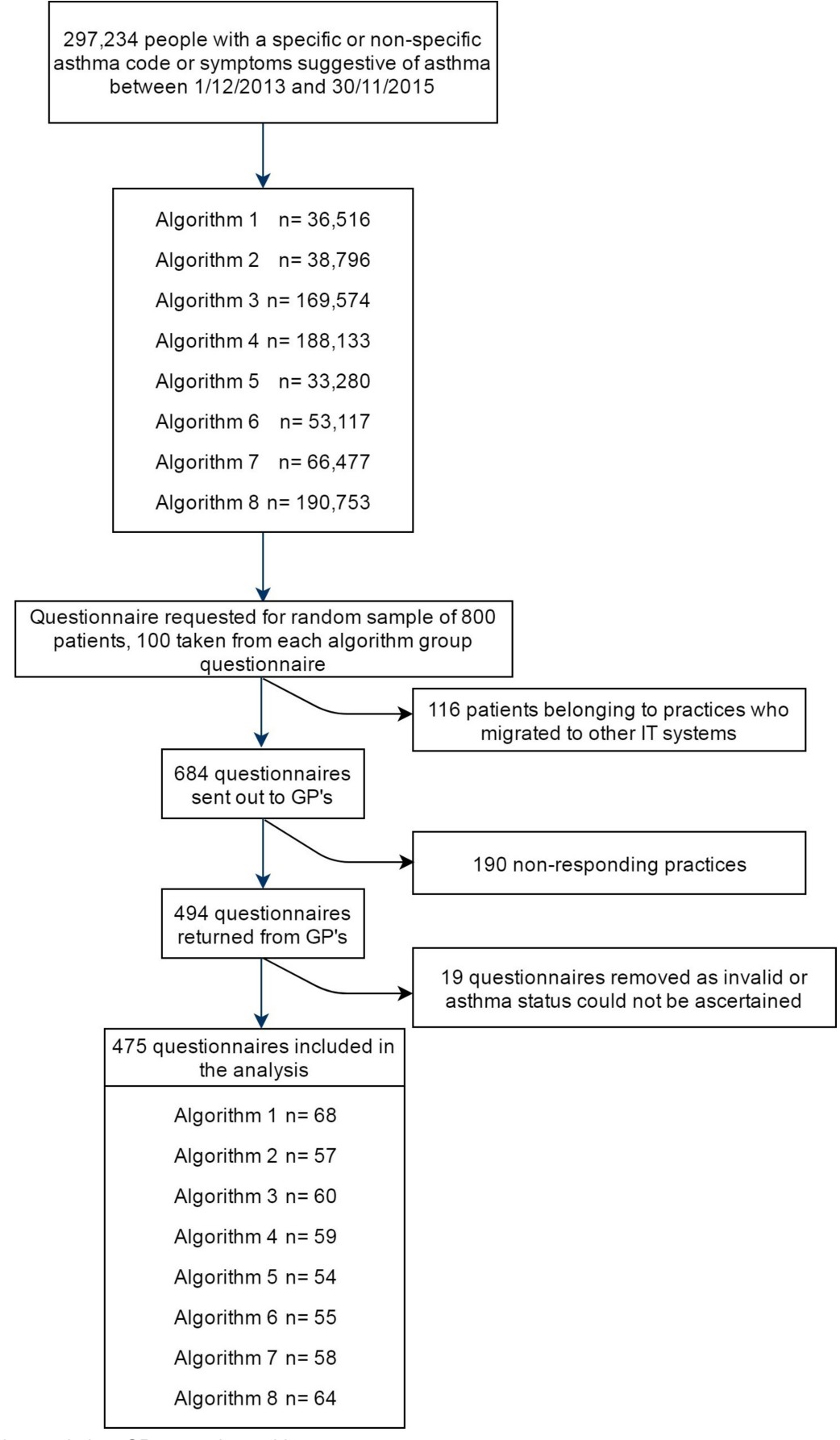

**Figure 1** Study population. GP, general practitioner.

make up the eight algorithms. The first four algorithms required a specific asthma diagnosis code, with the first three requiring additional documentation consisting of either respiratory medication use and/or evidence of reversibility testing. The fifth algorithm required a non-specific asthma code and additional documentation of both respiratory medications and reversibility testing; the last three algorithms required respiratory symptom codes indicating asthma symptoms with additional information. The presence of spirometry for inclusion in an algorithm was based on the existence of a specific spirometry Read code in the records rather than an examination of said spirometry, although where spirometry traces were provided as part of the additional information, they were examined. Evidence of reversibility testing only refers to whether airflow measurements or trial of treatment was done, and does not reflect the results of these tests. Respiratory medication use was defined as at least two prescriptions of asthma medication for inhaled asthma therapy (short-acting beta-agonists, long-acting beta-agonists and inhaled corticosteroids) within 365 days of each other, within the 2 years before the index date. From the expected most specific to most sensitive, the eight algorithms were constructed as follows:

1. Specific asthma Read code + evidence of reversibility testing (spirometry, variable peak expiratory flow rate or trial of treatment) + respiratory medications
2. Specific asthma code + evidence of reversibility testing
3. Specific asthma code + respiratory medications
4. Specific asthma code only
5. Non-specific asthma code + evidence of reversibility testing + respiratory medications
6. Asthma symptoms (wheeze, breathlessness, chest tightness, cough) + evidence of reversibility testing + respiratory medications
7. Asthma symptoms + evidence of reversibility testing
8. Asthma symptoms + respiratory medications

### Primary outcome
The primary outcome was confirmation of a diagnosis of asthma in each of the eight predefined algorithms. The gold standard for the diagnosis of asthma was the adjudicated asthma status agreed by the two study physicians, a respiratory physician and a GP who reviewed all questionnaires and evidence from the patient's GP independently. The reviewers were blinded to the code lists/algorithm. Where opinion differed, the cases were discussed and agreement was reached by consensus. The reviewing physicians did not know with which algorithm a person was selected.

### Statistical analysis
The PPV was calculated using the proportion of cases identified by each algorithm that were confirmed as actual cases by the study physicians through a review of the questionnaire and supporting evidence. All analyses were conducted using Stata V. 14.0.

A patient could contribute only to a single algorithm for the main analysis. In the post hoc analysis, individuals could be placed into multiple algorithms where possible to reduce the CI. The PPV in this analysis was calculated for all individuals who had a specific asthma code compared with those with a specific asthma code and additional information. We also performed a sensitivity analysis to check whether the age and sex for patients whose questionnaire was returned were similar to the age and sex of those patients whose questionnaire was not sent out or were there was no response. The study protocol is included in online supplementary appendix 3.

### Sample size calculation
As there were 116 patients that could not be evaluated, precision was expected to be slightly lower than in the original sample size calculations. However, a percentage difference in PPV of 0.13 is demonstrable with a sample size of 60 per algorithm (assuming PPV=0.85, $\alpha$=0.05 and power=0.8).

### RESULTS
A total of 800 potential asthma cases were selected for validation, of which 116 cases had migrated out of the database at the time the questionnaires were sent. Of the remaining 684 cases, there were 494 returned questionnaires. Nineteen of the returned questionnaires were considered invalid. Thus, 475 valid questionnaires were received, which yielded a response rate of 69.4% (475/684) using the practices that could have answered as denominator, as shown in figure 1. The time interval between the mailing of questionnaires and the review by the study physicians varied, but none of these time intervals was greater than 8 months.

The baseline characteristics of the 475 patients with valid returned questionnaires are shown in table 1. The study populations were mostly middle aged, never smokers and female. There were 97 individuals whose smoking status was not filled in on the questionnaire. Differences in the majority of characteristics were seen among most algorithms.

The PPVs of the eight algorithms are displayed in table 2.

The PPVs of algorithms containing specific or non-specific asthma codes in algorithms 1–5 (ranging from 83.3% to 90.7%) are markedly higher than the PPVs of the algorithms based on asthma symptoms (ranging from 43.1% to 57.8%). The combination of a specific code and asthma medication prescription and/or evidence of reversibility testing (PPV varies from 83.3% to 86.8%) did not considerably increase the PPV compared with a specific asthma code alone (PPV 86.4%). The highest PPV was found in the fifth algorithm combining a non-specific asthma code with evidence of reversibility testing and asthma medication use. However, the total number of patients identifiable with this algorithm (n=33 280) was less than one-fifth of those identifiable by the fourth algorithm

**Table 1** Characteristics of the 475 patients included in the final study analysis

| Algorithm | 1. Specific asthma code + reversibility testing + medication | 2. Specific asthma code + reversibility testing | 3. Specific asthma code + medication | 4. Specific asthma code | 5. Non-specific asthma code + reversibility testing + medication | 6. Symptoms + reversibility testing + medication | 7. Symptoms + reversibility testing | 8. Symptoms + medication | Total |
|---|---|---|---|---|---|---|---|---|---|
| Individuals, n (%) | 68 (100) | 57 (100) | 60 (100) | 59 (100) | 54 (100) | 55 (100) | 58 (100) | 64 (100) | 475 |
| Asthma diagnosis by patient's GP | 56 (82.4) | 49 (86) | 48 (80) | 51 (86.4) | 48 (88.9) | 29 (52.7) | 23 (39.7) | 38 (59.4) | 342 |
| Confirmation by respiratory physician before study start | 55 (80.9) | 29 (50.9) | 38 (63.3) | 45 (76.3) | 34 (63) | 23 (41.8) | 25 (43.1) | 36 (56.3) | 285 |
| Evidence of reversible airway obstruction | 47 (69.1) | 37 (64.9) | 32 (53.3) | 32 (54.2) | 31 (57.4) | 26 (47.3) | 19 (32.8) | 26 (40.6) | 250 |
| Mean age | 52.3 | 51.4 | 47 | 41.9 | 45 | 60.9 | 61.3 | 52.1 | |
| Mean age (95% CI) | (47.4 to 57.2) | (46.2 to 56.7) | (41.4 to 52.6) | (36.1 to 47.6) | (38.7 to 51.3) | (55.3 to 66.4) | (57.1 to 65.5) | (45.4 to 58.7) | |
| <18years old (%) | 7.35 | 7.02 | 15.25 | 18.64 | 16.67 | 7.27 | 1.72 | 20.31 | 11.81 |
| Sex: male | 31 (45.6) | 17 (29.8) | 16 (26.7) | 23 (39) | 26 (48.1) | 28 (50.9) | 24 (41.4) | 31 (48.4) | 196 |
| Current smoker* | 11 (16.2) | 10 (17.5) | 10 (16.7) | 5 (8.5) | 4 (7.4) | 5 (9.1) | 8 (13.8) | 4 (6.3) | 57 |
| Ex-smoker* | 16 (23.5) | 14 (24.6) | 17 (28.3) | 16 (27.1) | 15 (27.8) | 11 (20) | 10 (17.2) | 12 (18.8) | 111 |
| Never smoker* | 35 (51.5) | 26 (45.6) | 25 (41.7) | 36 (61.0) | 32 (59.3) | 18 (32.7) | 11 (19.0) | 27 (42.2) | 210 |
| Individuals with supporting info | 23 (33.8) | 21 (36.8) | 22 (36.7) | 14 (23.7) | 14 (25.9) | 17 (30.9) | 14 (24.1) | 22 (34.4) | 147 |

*As stated by patient's GP on the study questionnaire.
GP, general practitioner.

**Table 2** The positive predictive value (PPV) and proportion of patients diagnosed with chronic obstructive pulmonary disease within each algorithm

| Algorithm | Eligible population | Questionnaires sent out | Valid returned questionnaires (n, %) | Confirmed asthma cases | PPV (95% CI) |
|---|---|---|---|---|---|
| Specific asthma code + reversibility testing + medication | 36 516 | 92 | 68 (60) | 61 | 86.8 (78.5 to 95.0) |
| Specific asthma code + reversibility testing | 38 796 | 90 | 57 (63.3) | 51 | 86.0 (76.7 to 95.3) |
| Specific asthma code + medication | 169 574 | 89 | 60 (67.4) | 51 | 83.3 (73.6 to 93.0) |
| Specific asthma code | 188 133 | 84 | 59 (70.2) | 51 | 86.4 (77.4 to 95.4) |
| Non-specific asthma code + reversibility testing + medication | 33 280 | 78 | 54 (69.2) | 49 | 90.7 (82.8 to 98.7) |
| Symptoms + reversibility testing + medication | 53 117 | 87 | 55 (63.2) | 32 | 56.4 (42.8 to 69.9) |
| Symptoms + reversibility testing | 66 477 | 88 | 58 (65.9) | 26 | 43.1 (30.0 to 56.2) |
| Symptoms + medication | 190 753 | 78 | 64 (82.1) | 38 | 57.8 (45.4 to 70.2) |

Medication use was defined as two prescriptions within 365 days. Evidence of reversibility testing does not hold information on the outcome of these tests.

consisting of a specific asthma code alone (n=188 133) in the chosen time period. We have not examined the validity of a non-specific asthma code alone.

A post hoc analysis was performed where individuals were placed in every algorithm they qualified for. In this analysis, we found that the use of additional information on evidence of reversibility testing or medication in an algorithm with a specific asthma code again did not meaningfully increase the PPV. The PPV for all individuals who had a specific asthma code and information on reversibility testing or medication was 86.7% (95% CI 83.3% to 90.1%), and the PPV for individuals with only a specific asthma code was 86.4% (95% CI 83.0% to 89.7%).

When validating the record of possible asthma with a gold standard based on the study physicians' view of extra evidence provided by the GP, the PPV slightly improved across all algorithms. Figure 2 demonstrates the PPV of the different algorithms as diagnosed by the patient's own GP and the study physicians (overall κ=0.81).

There was no considerable difference in age or sex between patients whose questionnaire was returned and patients whose questionnaire was not sent out (age: p=0.74, sex: p=0.73) or were there was no response (age p=0.50, sex p=0.13) using $\chi^2$ tests.

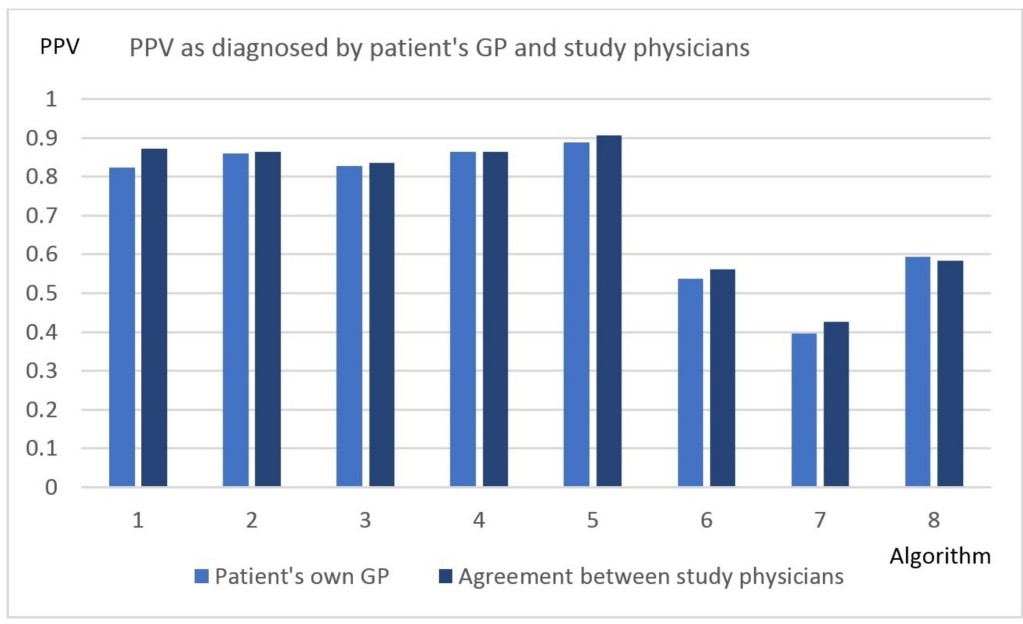

**Figure 2** PPV as diagnosed by the patient's own GP, and agreement between the study physicians. GP, general practitioner; PPV, positive predictive value.

## DISCUSSION

We tested the accuracy of eight algorithms to identify asthma within CPRD using a gold standard constructed using a consensus of the two study physicians. The algorithm with the highest PPV consisted of a combination for non-specific asthma codes, evidence of reversibility testing and multiple asthma prescriptions within 1 year (PPV 90.7, 95% CI 82.8 to 0.98.7) followed by a combination for specific asthma codes, evidence of reversibility testing and multiple asthma prescriptions within 1 year. The CI of this PPV overlaps with the CIs of each of the PPVs of the first four algorithms based on specific asthma codes, so the difference might be due to chance alone. The algorithm with the lowest PPV consisted of asthma symptoms and evidence of reversibility testing (PPV 0.43, 95% CI 0.30 to 0.55). The results of this validation study suggest that the clinical code-based algorithms that use asthma codes to identify asthma cases have high PPVs (between 0.84 and 0.91). In this dataset, a specific asthma code algorithm alone appears sufficient to identify current patients with asthma from CPRD. As the additional requirements of medication use and evidence of reversibility testing do not appear to significantly increase the PPV, the total number of individuals who can potentially be included in a study increases from 33 280 to 188 133 in the chosen time period (1 December 2013 to 30 November 2015). The total identifiable population of people living with asthma is thus much larger when only using a specific asthma code for identification.

### Comparison with previous studies

Validity of asthma codes in EHRs can be assessed by comparison to three different sets of gold standard: comparison to an external database, questionnaire and manual review by a clinician. This validation study uses questionnaires and manual review. Our gold standard consisted of the agreement of the study respiratory physician and study GP, both of whom were experienced with CPRD.

Previous studies which validated asthma in other EHR databases used manual review by clinicians to validate asthma in EHR and all reported at least one algorithm with a PPV above 85%.[21–26] In contrast with this study, the best results in previous studies arose when combining diagnostic data and prescription data.

The CPRD has provided anonymised primary care records for public health research since 1987; research was always a focus of interest when it was established. GPs contributing to the CPRD have been trained on how to record data for research use. As a consequence, data quality may be higher than in many other databases, in which research is only a secondary product.

### Strengths of this study

This study has several strengths. First, we were able to investigate the accuracy of eight predefined different algorithms and how they perform in identification of people with asthma in CPRD, as well as the accuracy of the actual GP diagnosis of asthma using additional information provided. Second, we included supporting information such as outpatient referral letters, other emergency department discharge letters, airflow measurements and radiography records. Finally, we validated asthma diagnoses found in CPRD, which is a primary care database that is extensively used for studying different health outcomes in epidemiological research. This primary care database provides health and medication history of millions of patients. A validated definition in CPRD of asthma allows for informed healthcare service planning by increasing the reliability of evidence generated from observational studies.

### Limitations of this study

This study has limitations to consider. The gold standard consisting of a GP questionnaire and review by study physicians is not absolute, even if we mitigated this with additional information from secondary care. A GP can look in the EHR to see if a specific diagnosis has been recorded for a specific patient when asked. This may lead to an overestimation of the PPV, but there is no suitable practical alternative. Ideally, airflow measurements and reversibility testing on each potential patient would form the optimal gold standard, but this would not be feasible in this setting due to cost. The overall number of questionnaires sent out (n=684) was less than requested (n=800) as some patients and practices were no longer part of CPRD and could not be contacted. However, the precision of PPV estimates was not substantially reduced.

Although practices contributing to CPRD are a sample of all practices in the UK, they are considered representative of the UK population with few patients opting out of contributing data, and is therefore unlikely to bias the results.[14]

GPs of patients with complicated medical histories could be less likely to return the questionnaire. The GPs were remunerated for their participation however, which is likely to have reduced the chance of this happening. Within the returned questionnaires, the amount of missing data was low, which suggests reasonable data quality. In addition, only living patients were assessed, as GPs no longer have access to the patient records after death. This excludes the records of the deceased patients and could result in survival bias. Patients had to be alive to be included, but it is unlikely that coding would differ between living and deceased individuals. If deceased people had died of asthma, the PPV in this study would be underestimated. Our findings are likely to be generalisable to other UK primary care databases using Read coding, but these would ideally still require validation. Databases using other coding systems may need to validate different algorithms to identify asthma, which might limit the generalisability of our findings. Another limitation is that we were not able to assess the negative predictive value of asthma diagnoses in CPRD because we evaluated only patients belonging to one of the eight algorithms. We could not calculate the specificity or sensitivity as we

had preselected our population of possible asthma cases. We also assumed the validity of asthma diagnoses would not be different between common and less frequent Read codes and the quality of recording would also be comparable for pragmatic reasons. However, the less commonly used codes will by definition identify a smaller proportion of all patients with asthma, so the validity we report will apply to the majority of patients.

## CONCLUSION

We have successfully estimated the PPV of several different algorithms to identify people with asthma in CPRD. The PPVs for specific asthma Read codes alone and non-specific ones in a combination with additional evidence were all greater than 0.84. A specific asthma code algorithm alone appears to be the most practical approach to identify patients with asthma in CPRD (PPV=0.86; 95% CI 0.77 to 0.95). Diagnoses were confirmed in a high proportion of patients with specific asthma codes, suggesting that epidemiological asthma research conducted using CPRD data can be conducted with reasonably high validity.

**Contributors** JKQ, IJD, LS and HM were responsible for developing the research question and have advised on the data collection and search strategies. FN summarised and analysed the questionnaires and drafted the manuscript. JKQ and DM reviewed the questionnaires and constructed the gold standard for asthma validation. JKQ is responsible for study management and coordination. All authors have read, commented on and approved the final manuscript.

**Funding** This work was supported by GlaxoSmithKline (GSK), through a PhD scholarship for FN with grant number EPNCZF5310. The publishing of this study was supported by the Wellcome Trust: grant number 098504/Z/12/Z.

**Competing interests** FN is funded by a GSK scholarship during his PhD programme. IJD is funded by an unrestricted grant from, has consulted for and holds stock in GlaxoSmithKline. HM is an employee of GSK R&D and own shares of GSK Plc. JKQ reports grants from MRC, BLF, Wellcome Trust and has received research funds from GSK, AZ, Quintiles IMS, in addition to personal fees from AZ, Chiesi, BI .

**Patient consent** This is a study using electronic health records; individual-level permission was given at the time of data collection. It does not need to be repeated for each study. All information is anonymised by CPRD.

**Ethics approval** Ethics approval was obtained from ISAC (the Independent Scientific Advisory Committee overseeing CPRD), protocol 15_257.

**Provenance and peer review** Not commissioned; externally peer reviewed.

**Data sharing statement** Study data will be available on request to FN once the research team has completed preplanned analyses.

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
