## [Reviewer comments · BMJ Open]

ARTICLE DETAILS

TITLE (PROVISIONAL)	Validation of asthma recording in the Clinical Practice Research Datalink (CPRD)
AUTHORS	Nissen, Francis; Morales, Daniel; Müllerová, Hana; Smeeth, Liam; Douglas, Ian; Quint, Jennifer

VERSION 1 - REVIEW

REVIEWER	Christoph R. Meier Basel Pharmacoepidemiology Unit Department of Pharmaceutical Sciences University of Basel Switzerland
REVIEW RETURNED	12-May-2017

GENERAL COMMENTS	Nissen et al. conducted a validation study to explore the validity of asthma diagnoses in the CPRD. They defined at random a sample of patients with evidence for asthma and assessed the PPV for 8 different algorithms. They concluded that overall the validity of asthma diagnoses is high, as long as searching for them is based on asthma codes and not only symptoms. Overall, the paper is nicely written and clear, and it adds to the body of validation studies establishing the CPRD as great and highly valid tool for epidemiological research. I have only a few comments: Methods: the index date is Dec 2015, and the authors searched for codes within the preceding two years. They nicely explain why they did this, and it makes sense. However, they may want to address an issue that is often a problematic point in the CPRD: some GP record a code only once in the record of a patient; if the disease is chronic and remains, this one recording is usually enough to classify a patient as prevalent case. Other GPs keep recording even chronic diagnoses over and over again, for example whenever they see the patient and ask about how the patient is currently with regard to this disease X. In other words, patients with an asthma code within two years prior to the index date may indeed have had a first recording of asthma during this predefined time period, or they may have already have had a long-term history and a repeated recording of asthma. This may not affect the validity of their findings, and maybe there is also a lack of power, but a stratified look at those with incident asthma vs. those with a long-term history may be interesting, if feasible. Methods: did the authors have a look at the timing of the recording of asthma with regard to seasonality? Asthma is often a complication of allergic rhinitis in spring, and these transient cases may differ from those with chronic disease. Did the authors look at the PPV by season? Maybe a stratification of asthma into 'spring vs. the rest of
---

	the year' might also be of interest. Discussion: The authors refer to databases containing electronic health records (HER) as a whole, as if they were all the same. In reality, there are substantial differences in quality and comprehensiveness of recorded data across databases. Claims data for example have their own problems, and data collections based on particular hospitals, areas or special populations may also be prone to biases and quality issues. The CPRD has been established some 30 years with a clear emphasis on research; although it is the tool in daily practice for GPs to manage their patients, research was always a focus of interest when it was established. This resulted in a higher data quality than in many other databases, in which research is only a by-product. In the UK, GPs have been trained how to record data appropriately in the CPRD. This fact may merit a sentence or two in the Discussion.
--	---

REVIEWER	Joseph A. Pacheco The University of Kansas Medical Center, U.S.A.
REVIEW RETURNED	30-May-2017

GENERAL COMMENTS	This is a very interesting paper. I have a few minor issues that can be easily cleared up. The way the article is introduced shows asthma as a worldwide problem, but the article focuses on the UK population exclusively. There should be some mention of how many people are affected by asthma in the UK, to better set up why this article is important. EHRs have been adopted worldwide, but there is still an issue compatibility between EHRs. This compatibility may be a non-issue in the UK due to the NHS and are the proposed algorithms? If not this should be added to the limitations. There was one copy/edit issue I noticed. On page 12 line 44 you have "post hoc" not hyphenated (correct), and on page 14 line 46 you have "post-hoc" hyphenated (not correct). Other than these minor issues great article.
--

VERSION 1 – AUTHOR RESPONSE

Reviewer: 1

Reviewer Name: Christoph R. Meier

Thank you for the suggestions, we have explored the possibility to look at the validity of incident versus chronic asthma codes and the validity of asthma seasonality.

A stratified look at those with a single asthma code versus those with a longer history of asthma did not prove feasible due to a lack of power and a wide variation of possible follow-up time. For example, the follow-up of some patients with an asthma code is less than 6 months, but we cannot whether they had received a previous asthma diagnosis in a practice not contributing to the CPRD. Some codes are more likely represent acute asthma (eg "severe asthma attack"), while others are more likely to be chronic (eg "asthma monitoring plan given"), which could also influence validity. However, we are testing the code list as a whole, without looking to specific codes. Ultimately, this paper focuses on the validity of at least one recording of a specific or non-specific asthma code. The issue of validity of incident versus chronic disease is beyond the scope of this paper.

Regarding seasonality: We had selected our algorithm populations using people with any number of

asthma codes in the two years before index date. This means that we do not have one specific asthma coding with a date for people who have had more than one asthma code in these two years. Thus, looking at seasonality is be unfeasible, and would introduce bias if we would only look at (for example) the last recorded asthma code.

Discussion: We have included a few sentences on the research purpose and history of the CPRD in the discussion section.

Reviewer: 2

Reviewer Name: Joseph A. Pacheco

We have added the epidemiology of asthma in specifically the UK in the background section.

Regarding compatibility: CPRD can be linked to HES (Hospital Episode Statistics) for data on hospital stays and A&E visits, and to ONS (Office of National Statistics) for data on mortality. Compatibility issues between databases exist, but are solvable. Not all NHS practices contribute to the CPRD, but the contributing practices are nonetheless considered to be representative of the UK population. The spelling error "post-hoc" has been corrected.

VERSION 2 – REVIEW

REVIEWER	Christoph R. Meier Basel Pharmacoepidemiology Unit Department of Pharmaceutical Sciences University of Basel , Switzerland
REVIEW RETURNED	30-Jun-2017

GENERAL COMMENTS	The authors addressed the points which have been raised adequately.
---

REVIEWER	Joseph A. Pacheco University of Kansas Medical Center, U.S.A.
REVIEW RETURNED	08-Jul-2017

GENERAL COMMENTS	This is a great revision. I believe that this manuscript adds to field of research, especially with use of EHRs.
--